# SMART: A Small-to-large Model Coordinated Retrieval and Response Framework for Task-Oriented Dialogue

## Abstract

Task-Oriented Dialogue (TOD) systems are commonly used to assist users in achieving specific goals through human-computer interactions. Existing methods typically employ a single model during response generation to simultaneously learn response policy and perform fine-grained knowledge retrieval. However, these task-coupling methods often lead to suboptimal response generation, incorporating irrelevant knowledge and policy-violating styles, which makes optimizing TOD systems more difficult. To address this challenge, we propose a novel small-to-large model collaboration framework for task-oriented dialogue, named SMART. This framework utilizes a small language model to generate style-only responses without knowledge, while a large language model retrieves top-1 relevant knowledge from the knowledge base independently. The style-only responses are finally filled with top-1 relevant knowledge and form the completed responses to users. Finally, a re-thinker is designed to check the contextual relevance and knowledge accuracy of responses. Experiments on two public datasets demonstrate that SMART outperforms existing methods by an average of 5.79% in Entity F1.

## 1 Introduction

Task-Oriented Dialogue (TOD) systems interact with users in natural language to help them accomplish information-driven tasks, such as hotel booking, weather inquiry, and scheduling tasks Chen et al. (2017). All of these tasks require TOD systems to generate highly accurate knowledge-based responses and perform specific actions. Therefore, TOD tasks often necessitate multi-step knowledge retrieval from an external knowledge base to generate accurate responses for users Qin et al. (2023a). As illustrated in Figure 1 (a), some existing methods directly utilize a single language model (LM) to generate responses based on both the dialogue context and the entire knowledge base Rony et al. (2022); Dong & Chen (2023). In these methods, the LM-based generators produce responses token by token while simultaneously performing knowledge retrieval. However, there are two main challenges: (1) When the knowledge base is large, the model input becomes excessively long, leading to an increased response generation time. (2) Since an entire knowledge base is used as input, the model struggles to generate precise responses among a vast number of knowledge records.

To address these challenges, some works propose alleviation strategies Wan et al. (2023); Shi et al. (2023); Shen et al. (2023). These methods employ a knowledge retriever to first select the top-$K$ relevant knowledge records from the knowledge base, and then feed the selected records together with the dialogue context to the generator for response generation Chen et al. (2025a), as shown in Figure 1 (b). However, they still require the LM-based generator to retrieve fine-grained knowledge from the top-$K$ records during response generation. In practice, these methods frequently generate suboptimal responses that include policy-violating styles and irrelevant knowledge. Another key limitation of these approaches lies in their retrievers: they typically encode each entire knowledge record into a single vector and perform retrieval based on vector similarity. This coarse-grained matching strategy makes it difficult to capture fine-grained attribute-value differences between knowledge records. As a result, the retrieved top-$K$ knowledge may still contain irrelevant or noisy entries.

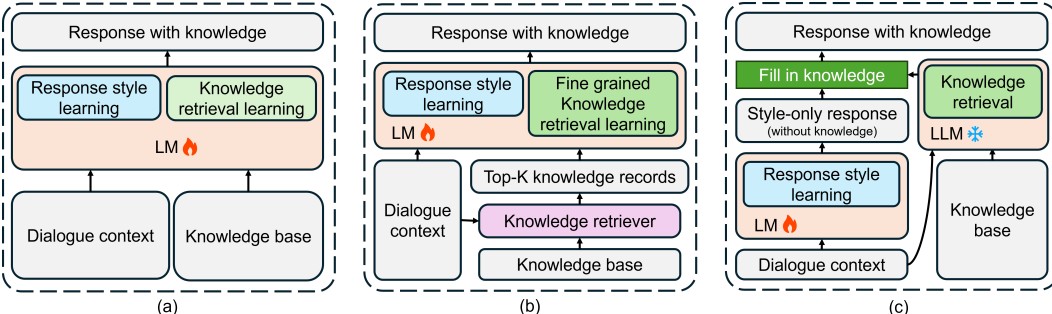

Figure 1: Methods for task-oriented dialogue. (a) This kind of method employs the dialogue context and entire knowledge base as input to LM for response generation with knowledge. (b) This kind of method utilizes a knowledge retriever to conduct coarse-grained knowledge retrieval. (c) Our method, which totally decouples response generation and knowledge retrieval, enabling each component to focus on a single task.

In this paper, we introduce SMART (**S**mall-to-large **M**odel coll**A**boration f**R**amework for **T**ask-oriented dialogue), a novel framework designed to alleviate the task-coupling limitations of existing TOD systems. SMART fully decouples knowledge retrieval from response generation, enabling each component to focus on a single task. A Small Language Model (SLM) is trained to generate style-only responses containing placeholder slots for attributes, while a training-free Large Language Model (LLM) leverages a fine-to-coarse retrieval mechanism to retrieve the most relevant knowledge record (top-1) from an attribute-filtered knowledge base which filters irrelevant attributes based on the attribute slots. Then, the attributes of the top-1 knowledge record are directly filled into the slots of the style-only response. This task separation allows for targeted optimization and better generalization. To ensure a relevant and accurate final response, we introduce a re-thinker module that evaluates both the contextual relevance and correctness of the generated output. Experiments on two public datasets demonstrate that SMART consistently outperforms existing methods, achieving an average improvement of 5.79% in Entity F1 score. In summary, this paper provides three contributions:

- We propose a new small-to-large model collaboration framework for task-oriented dialogue named SMART. It synergizes the efficiency of a small language model for response style learning with the precise knowledge retrieval of a frozen large language model, and thereby enables a balanced and adaptable method for task-oriented dialogue.

- We introduce a novel fine-to-coarse knowledge retrieval mechanism. It leverages attribute slots from style-only responses to filter out irrelevant knowledge attributes, thereby narrowing the retrieval scope to select the top-1 knowledge record. This mechanism improves the precision of retrieving relevant knowledge while reducing computational costs.

- Experiments on all two publicly available datasets demonstrate our proposed method outperforms state-of-the-art baselines. A series of in-depth experiments analyses validates the effectiveness of both the small-to-large model collaboration framework and the fine-to-coarse knowledge retrieval mechanism.

## 2 RELATED WORK

Existing end-to-end task-oriented dialogue (TOD) systems can generally be categorized into two paradigms: the unified paradigm and the pseudo-pipeline paradigm.

The unified paradigm employs a single model to directly generate responses based on both the dialogue context and the knowledge base. For instance, Madotto et al. (2018) propose Mem2Seq, an end-to-end neural network that uses a memory network to store dialogue context and knowledge records, followed by a pointer network to generate responses. Qin et al. (2020) extend Mem2Seq with a shared-private framework to separately learn domain-shared and domain-specific knowledge. With the rise of pre-trained language models (PLMs), many works adopt PLMs to build unified systems. Budzianowski & Vulić (2019) propose fine-tuning PLMs for dialogue tasks, while Yang et al. (2021) directly use GPT-2 to model the entire dialogue session as a sequence of user utterances,

belief states, database results, system acts, and responses. Wang et al. (2022) enhance GPT-2 with lightweight adapter and CopyNet modules (GPT-Adapter-CopyNet) for better domain adaptation and entity generation. Rony et al. (2022) propose DialoKG, which dynamically embeds knowledge into context for GPT-2. To further improve generation accuracy, Ding et al. (2024) introduce a prefix trie to constrain GPT-2's outputs, ensuring consistency with the knowledge base.

However, jointly optimizing knowledge retrieval and response generation in a single model often leads to suboptimal performance and poor interpretability. To address this, the pseudo-pipeline paradigm separates the TOD task into two loosely coupled components: a retriever and a generator. For example, Cai et al. (2019) utilize a retriever to retrieve past similar dialogues and extract a retrieval-based skeleton, which is then used to generate a final response. Tian et al. (2022) propose Q-TOD, which extracts query representations from dialogue context for knowledge retrieval, followed by response generation. Wan et al. (2023) propose MAKER, a multi-granular retriever that includes both entity and attribute selectors to improve knowledge relevance. Shen et al. (2023) use maximal marginal likelihood to train the retriever using feedback from response generation. Similarly, Shi et al. (2023) propose a dual feedback mechanism to jointly optimize the retriever and generator. Dong et al. (2024) employ LLMs as knowledge retrievers and separate LMs for generation. SAGE Chen et al. (2025a) further introduces span-level attention in the generator to focus on a single knowledge record among top-K retrieved candidates, alleviating entity inconsistency.

In summary, while recent pseudo-pipeline methods such as MAKER and SAGE attempt to separate knowledge retrieval and response generation, they still rely on a single model to perform each sub-task. In contrast, our SMART framework explicitly coordinates a small model for style-only response generation and a large model for knowledge retrieval and verification, enabling a clear functional separation and enhanced interpretability.

## 3 METHOD

### 3.1 PROBLEM DEFINITION

For the $i$-th dialogue turn, the dialogue context $d_i$ consists of the full dialogue history and the current user utterance $u_i$. It is defined as $d_i = [u_1, s_1, \ldots, u_{i-1}, s_{i-1}, u_i]$, where $u_j$ and $s_j$ denote the user utterance and system response at turn $j$. An external knowledge base $K$ is used to provide knowledge for TOD system to generate a informative response. The knowledge base contains $N$ knowledge records, denoted as $K = [e_1, e_2, \ldots, e_N]$. Each knowledge record $e_i$ is a collection of attribute–value pairs and can be represented as $e_i = [(a_i^1, v_i^1), (a_i^2, v_i^2), \ldots, (a_i^j, v_i^j)]$, where $a_i^j$ denotes the $j$-th attribute and $v_i^j$ denotes its corresponding value. The attribute set of the knowledge base is denoted as $A = \{a^1, a^2, \ldots, a^M\}$, where $M$ is the total number of attributes in knowledge base schema. The system generates a response $s_i$ based on the $i$-th turn dialogue context $d_i$ and the knowledge base $K$.

### 3.2 MODEL ARCHITECTURE

The proposed SMART consists of a response style generator, a fine-to-coarse knowledge retriever, and a re-thinker. The overall architecture of SMART is shown in Figure 2. Firstly, the response style generator according to the $i$-th turn dialogue context $d_i$ to generate the style-only response $\hat{p}_i$. This response is not a completed response, there are some placeholder attribute slots $A_i^{'}$ in it. These attribute slots are then used as attribute filters for fine-to-coarse knowledge retrieval. That means for each record in the knowledge base, only the attributes corresponding to the slots are retained for subsequent top-1 knowledge retrieval, while the remaining attributes are removed by an attribute filter. Next, the parameters-frozen LLM-based knowledge retriever is utilized to retrieve the most relevant knowledge based on dialogue context $d_i$ and an attribute-filtered knowledge base $K^{'}$. The values of the attributes from the retrieved knowledge are finally filled into the style-only response and form the $i$-th turn completed response $\hat{s}_i$.

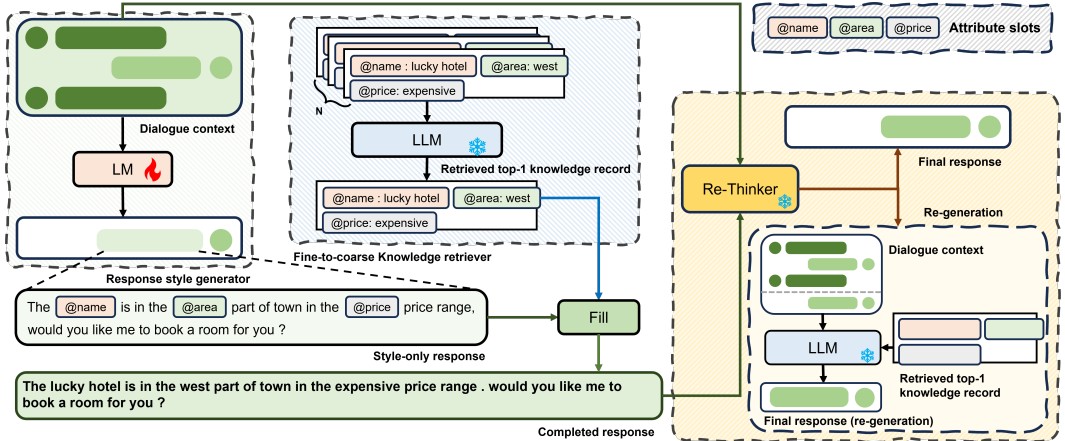

Figure 2: The overall architecture of SMART.

### 3.3 THE RESPONSE STYLE GENERATOR

The response style generator is designed to rapidly generate a style-only response that includes specific attribute slots $A_i'$, which are essential for both fine-to-coarse retrieval and final response generation. Given the $i$-th dialogue context $d_i$, an encoder $Enc_d$ transforms the raw dialogue context into a vector representation $h_i^{ctx}$. As $Enc_d$ consists of an SLM, the context length is limited. we adopt a time-window strategy that retains only the most recent $K$ dialogue turns, truncating the full dialogue context $d_i$ as truncated context $d_i^{trn}$, and obtain a corresponding vector representation $h_i^{ctx'}$. The vector $h_i^{ctx'}$ is then passed to a decoder $Dec_d$ with a specific adapter layer, which decodes the embedded context into a predicted style-only response $\hat{p}_i$. This response is not fully completed and instead contains attribute slots $A_i'$, placeholders for specific details that will be filled after knowledge retrieval. Formally,

$$\hat{p}_i = LM(d_i^{trn}, \theta_{lm}, \theta_{ap}), \tag{1}$$

where $\theta_{lm}$ are the parameters of the SLM used for response style generator, $\theta_{ap}$ are parameters of the specific adapter layer.

To comprehensively train the response style generator, we adopt a multi-task training strategy. In addition to generating style-only responses denoted as $\hat{p}_i$, the model is also tasked with predicting the corresponding completed response $\hat{s}_i$. This auxiliary task encourages the model not only to capture the stylistic policy of responses but also to develop a better understanding of fine-grained knowledge usage, such as assessing whether specific attribute values are appropriate in context. To facilitate this, we introduce a completed response task adapter layer with parameters $\theta_{af}$ into the decoder $Dec_d$, enhancing the generator's versatility and its ability to generalize across varying dialogue contexts. Formally,

$$\hat{s}_i' = LM(d_i^{trn}, K, \theta_{lm}, \theta_{af}). \tag{2}$$

It is worth noting that the completed response generation task is only used in the training phase.

To train these two types of responses, we employ a straightforward yet effective approach using cross-entropy (CE) loss. The specific loss function used in the training process is described in Equation 3.

$$\mathcal{L}_{total} = CE(\hat{p}_i, p_i) + CE(\hat{s}_i', s_i) \tag{3}$$

The $p_i$ is the golden style-only response and $s_i$ is the golden completed response from the datasets.

### 3.4 THE FINE-TO-COARSE KNOWLEDGE RETRIEVER

To enable more precise knowledge retrieval, we propose a fine-to-coarse retrieval mechanism that incorporates attribute-based filtering. Each knowledge record in $K$ is associated with multiple attributes (e.g., "name", "phone", "address", and "type"). However, in most real-world scenarios, many attributes in the retrieved knowledge record are redundant and not ultimately used in the final response. To address this issue, we extract a subset of relevant attribute slots $A_i^{'}$ obtained from the style-only response to retain only the necessary information and discard irrelevant attributes from the knowledge base. The attribute-based filtering process is formally defined in Equation 4.

$$K^{'} = \text{Filter}(K, A_i^{'}) \tag{4}$$

$K^{'}$ is a fine-grained attribute knowledge base obtained by applying a simple filtering function $\text{Filter}$ to $K$, which retains only the attributes specified in $A_i^{'}$ for each knowledge record. Next, the fine-to-coarse knowledge retriever takes both the filtered knowledge base $K^{'}$ and the dialogue context $d_i$ as input. The retriever leverages the inference capabilities of LLM to assess the relevance of each candidate knowledge record, evaluating how well it matches the dialogue context. The used prompt $P^r$ designed as "*Dialogue context: {}. Knowledge records: {}. According to the dialogue context, determine which knowledge records are appropriate. Answering with the name of the knowledge record is fine.*". Through this process, the retriever directly selects the top-1 most relevant knowledge record for the $i$-th turn $e_i^{top}$, ensuring the most accurate response is generated in response generation phase. Formally,

$$e_i^{top} = \text{LLM}(d_i, K^{'}, P^r). \tag{5}$$

Given top-1 most relevant knowledge record $e_i^{top}$, the value of the attributes from the retrieved top-1 relevant knowledge record are filled into the style-only response and form the $i$-th turn completed response $\hat{s}_i$. Formally,

$$\hat{s}_i = \text{Fill}(e_i^{top}, \hat{p}_i) \tag{6}$$

### 3.5 THE RE-THINKER

The re-thinker is designed to reassess and refine the completed response by given the dialogue context $d_i$ and retrieved top-1 knowledge record. The re-thinker evaluates this completed response by concatenating it with the dialogue history to assess whether the response is appropriate in the given context. The prompt used for re-think is designed as "*Dialogue context: {}. A response: {}. Do you think the response fits the context? Answer yes or no is enough. Think step by step.*"

If the response is deemed appropriate, this completed response is used as the final completed response. Otherwise, a re-generation prompts $P^g$ is designed to guide the LLM to generate a contextually response. The prompt is "*Dialogue context: {}. Knowledge records: {}. An example of a non-applicable response: {}. You need to respond to the user based on the given dialogue context, the non-applicable response sample, and the knowledge record I have provided to you. So, your response is?*"

This re-thinker ensures that SMART output is both contextually knowledge relevant and style-compliant, while also being refined through an additional layer of evaluation to guarantee its overall quality and speed.

## 4 EXPERIMENT

### 4.1 DATASET

The effectiveness of the proposed model was validated using two publicly available task-oriented dialogue datasets: CamRest Wen et al. (2017), and MultiWOZ 2.1 (MWOZ) Budzianowski et al. (2018). The Camrest dataset primarily focuses on the restaurant reservation domain and contains

676 multi-turn dialogues. The MWOZ dataset consisted of five domains: train, hotel, restaurant, taxi, and attraction. It is composed of 2,877 multi-turn dialogues. Each dataset was further divided into training, validation, and test sets following previous work Wan et al. (2023).

## 4.2 EVALUATION METRICS

To evaluate model performance, we use five metrics: BLEU (4-gram), Entity F1, Entity precision, Entity recall, and Entity accuracy. BLEU Papineni et al. (2002) measures fluency via n-gram overlap between the enerated text and the reference text. Entity F1 Eric et al. (2017), the harmonic mean of precision and recall, assesses accurate entity identification and use in dialogue. Entity Precision evaluates the correctness of generated entities, Entity Recall measures coverage of reference entities, and Entity Accuracy gauges overall entity recognition accuracy.

## 4.3 BASELINE

To objectively evaluate our method, we select a variety of representative baseline models for comparison, categorizing them according to their knowledge retrieval strategies in response generation[1].

Implicit retrieval: These methods integrated the processes of knowledge retrieval and response generation within a single model, including GraphMemDialog Wu et al. (2022), ECO Huang et al. (2022), DialoKG Rony et al. (2022), MPEToDs Qin et al. (2023b), PluDG Dong & Chen (2023), Uni-ToD Ding et al. (2024) and IEM Chen et al. (2024).

Explicit retrieval: These methods utilize a knowledge retriever to retrieved the top-$K$ relevant knowledge records, only conduct fine-grained knowledge retrieval during the response generation, including Q-TOD Tian et al. (2022), MAKER Wan et al. (2023), DF-TOD Shi et al. (2023), RGA-TOD Shen et al. (2023), and MLTOD Dong et al. (2024).

## 4.4 IMPLEMENTATION

We use the base version of Flan-T5 Raffel et al. (2020) as the response style generator. During training, we set the learning rate to $5 \times 10^{-5}$ and employ the AdamW optimizer. Additionally, we utilize a Llama-3-8B model with frozen parameters as both the fine-to-coarse knowledge retriever and the re-thinker. All experiments are conducted on an NVIDIA RTX 3090 GPU with 24GB of memory.

## 4.5 MAIN RESULTS

To comprehensively compare the performance of SMART with several baseline models, we utilize two public datasets and five metrics to evaluate these methods. SMART consistently outperforms baseline models in all metrics across the datasets and achieved state-of-the-art (SOTA). On the Cam-Rest dataset, SMART achieves a BLEU score of 27.21, which represents a 4.7% improvement over the second-best baseline, IEM (25.99). In terms of entity metrics, SMART surpass baselines by 2.5% in Entity F1, 1.3% in Entity Precision, 3.1% in Entity Recall, and 3.9% in Entity Accuracy, respectively. On the MWOZ dataset, SMART achieves a BLEU of 18.82, improving by 3.1% over the previous SOTA baseline DF-TOD (18.26). Additionally, SMART outperforms LQ-TOD by 9.1% in Entity F1, 10.1% in Entity recall, and 5.9% in Entity accuracy. compared to MAKER, SMART achieves a 4.2% boost in Entity Precision. These results demonstrate SMART has an advantage in generating fluent, and knowledge-accurate response, highlighting effectiveness of our small-to-large model coordinated retrieval and response framework.

## 4.6 ABLATION STUDY

To explore the contribution of each part from SMART, we conduct the ablation study experiment[2]. Table 2 presents the results of the experiment. When we remove the fine-to-coarse knowledge retriever (*w/o fine-to-coarse knowledge retriever*) and use the coarse-to-fine knowledge retriever

---

[1]The descriptions of these baselines are detailed in the Section 2.

[2]The result of ablation study on CamRest dataset can be found in Appendix B.

Table 1: Performance comparison of SMART and baseline models on two public datasets (CamRest and MWOZ). Bold numbers indicate the state-of-the-art results. Numbers with underline indicate the second results.

| Dataset | Method | BLEU | Entity F1 | Entity Precision | Entity Recall | Entity Accuracy |
|---|---|---|---|---|---|---|
| CamRest | GraphMemDialog Wu et al. (2022) | 22.30 | 64.40 | - | - | - |
| | ECO Huang et al. (2022) | 18.42 | 71.56 | - | - | - |
| | DialoKG Rony et al. (2022) | 23.40 | 75.60 | 77.31 | 73.97 | 50.32 |
| | MPEToDs Qin et al. (2023b) | 19.30 | 58.90 | - | - | - |
| | PluDG Dong & Chen (2023) | 23.00 | 76.90 | 77.20 | 76.61 | 51.86 |
| | MAKER Wan et al. (2023) | 25.04 | 73.09 | 73.89 | 72.31 | 46.32 |
| | DF-TOD Shi et al. (2023) | 25.85 | 72.83 | - | - | - |
| | RGA-TOD Shen et al. (2023) | 25.00 | 72.09 | - | - | - |
| | Uni-TOD Ding et al. (2024) | 24.70 | 77.80 | 78.89 | 76.78 | 52.39 |
| | IEM Chen et al. (2024) | 25.99 | 77.12 | 77.34 | 76.91 | 51.05 |
| | MLTOD Dong et al. (2024) | 25.20 | 77.49 | 77.85 | 77.13 | 51.79 |
| | SMART(Ours) | **27.21** | **79.72** | **79.92** | **79.53** | **54.43** |
| MWOZ | GraphMemDialog Wu et al. (2022) | 14.90 | 40.20 | - | - | - |
| | ECO Huang et al. (2022) | 12.61 | 40.87 | - | - | - |
| | DialoKG Rony et al. (2022) | 12.60 | 43.50 | 43.87 | 43.12 | 31.47 |
| | MPEToDs Qin et al. (2023b) | 13.60 | 36.60 | - | - | - |
| | PluDG Dong & Chen (2023) | 9.20 | 42.40 | 43.56 | 41.32 | 30.28 |
| | MAKER Wan et al. (2023) | 17.23 | 53.68 | 55.17 | 52.27 | 37.21 |
| | DF-TOD Shi et al. (2023) | 18.26 | 52.52 | - | - | - |
| | RGA-TOD Shen et al. (2023) | 16.97 | 51.99 | - | - | - |
| | Uni-TOD Ding et al. (2024) | 12.30 | 44.30 | 46.27 | 42.49 | 33.07 |
| | IEM Chen et al. (2024) | 15.18 | 45.20 | 47.42 | 43.18 | 32.89 |
| | LQ-TOD Chen et al. (2025b) | 17.97 | 54.35 | 54.53 | 54.17 | 38.78 |
| | SMART(Ours) | **18.82** | **58.55** | **57.48** | **59.65** | **41.07** |

Table 2: Model ablation experiments for two different modules on MWOZ dataset.

| Method | BLEU | Entity F1 | Entity Precision | Entity Recall | Entity Acc. |
|---|---|---|---|---|---|
| Ours | **18.82** | **58.55** | **57.48** | **59.65** | **41.07** |
| w/o fine-to-coarse knowledge retriever | 14.34 (4.48↓) | 55.64 (2.91↓) | 54.47 (3.01↓) | 56.87 (2.78↓) | 39.22 (1.85↓) |
| w/o completed response generation task | 16.32 (2.50↓) | 56.32 (2.23↓) | 55.72 (1.76↓) | 56.93 (2.72↓) | 39.93 (1.14↓) |
| w/o re-thinker | 17.01 (1.81↓) | 57.31 (1.24↓) | 56.76 (0.72↓) | 57.87 (1.78↓) | 40.75 (0.32↓) |

from MAKER Wan et al. (2023) to conduct knowledge retrieval, the performance is depressing. As the same, when we remove the re-thinker (*w/o re-thinker*), SMART also has a performance decline. In addition, removing the completed response generation task (*w/o completed response generation task*) also leads to a noticeable drop across most evaluation metrics, showing that this module is essential for ensuring accurate and coherent response generation. These results indicate that the fine-to-coarse knowledge retriever, the re-thinker, and the completed response generation task all play crucial roles in the overall effectiveness of SMART, as their removal negatively impacts performance across all evaluation metrics.

## 4.7 EFFECT OF DIRECT LLM-BASED GENERATION

A potential concern is whether it is necessary to use an SLM for style-only response generation. One might argue that directly feeding the retrieved knowledge into an LLM for generation, with appropriate prompting or few-shot learning, could achieve similar effects without additional modeling. To verify this, we conduct experiments on the CamRest and MWOZ datasets. Although LLMs have strong reasoning ability, directly using them for response generation without finetuning often results in verbose outputs containing irrelevant or excessive attributes. This substantially lowers BLEU and Entity F1 scores. In contrast, training a style-only SLM ensures concise, stylized responses that are easier to control. Table 3 shows the results. Across both datasets, the direct LLM for generation underperforms compared to our style-only approach. While finetuning improves performance (e.g., Llama3-8B (Finetuned)), it still falls significantly short of our method. These

Table 3: Comparison between direct LLM Re-Thinker and our style-only LM approach on CamRest and MWOZ datasets.

| Method | CamRest | | MWOZ | |
|---|---|---|---|---|
| | BLEU | Entity F1 | BLEU | Entity F1 |
| Gemini-2 flash | 1.32 | 46.28 | 2.12 | 29.48 |
| GPT-4o | 2.19 | 47.91 | 1.43 | 29.90 |
| Qwen2.5-7B | 0.92 | 52.81 | 1.32 | 35.52 |
| Llama3-8B | 0.81 | 51.60 | 1.71 | 34.36 |
| Llama3-8B (Finetuned) | 5.38 | 54.87 | 2.45 | 35.32 |
| **Ours** | **27.21** | **79.72** | **18.82** | **58.55** |

Table 4: The impact of large-small language model collaboration framework.

| Retrieval method | Generation method | BLEU ↑ | Entity F1 ↑ | Time (s) ↓ |
|---|---|---|---|---|
| LLM Coarse-only Retrieval | LLM Generation | 7.41 | 41.39 | 16.98 |
| LLM Coarse to Fine Retrieval | LLM Generation | 9.74 | 43.09 | 23.67 |
| LLM Fine to Coarse Retrieval | LLM Generation | 10.34 | 47.05 | 14.32 |
| LLM Coarse-only Retrieval | LM Generation | 14.64 | 51.95 | 12.52 |
| LLM Coarse to Fine Retrieval | LM Generation | 16.49 | 53.95 | 16.38 |
| LLM Fine to Coarse Retrieval | LM Generation | **18.82** | **58.55** | 11.13 |
| LM Retrieval | LM Generation | 17.23 | 53.68 | **10.89** |

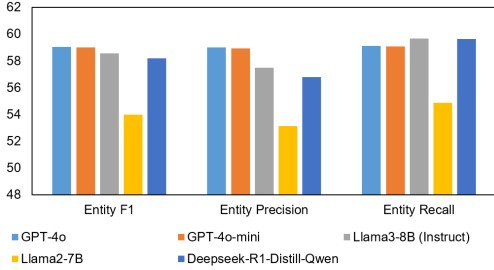

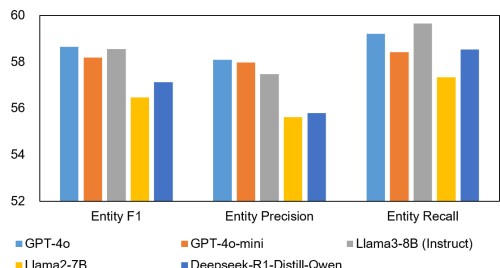

Figure 3: Different LLMs Performance in fine-to-coarse retrieval.

Figure 4: Performance of different LLMs in the re-thinker.

results confirm that the style-only SLM plays a crucial role in achieving high-quality task-oriented responses while costing fewer computational resources.

### 4.8 COORDINATED STRATEGY OF LARGE-SMALL LANGUAGE MODEL COORDINATED FRAMEWORK

In order to explore how to coordinate large language model and small language model to achieve the best performance for TOD, we try several model-coordinated strategies. Table 4 presents an analysis of different model-coordinated strategies between large and small language models within the coordinated framework of SMART. The table compares various retrieval methods: LLM based coarse-only retrieval, LLM based coarse-to-fine retrieval, and LLM based fine-to-coarse retrieval and their corresponding generation methods. We evaluate their impact on three metrics: BLEU, Entity F1, and inference time. The results reveal that the best-performing configuration is the strategy of LLM-based fine-to-coarse retrieval with LM-based generation, achieving the highest BLEU score (18.82) and Entity F1 (58.55). Moreover, regardless of the retrieval method, using LLM generation yields lower BLEU and Entity F1 scores than using LM generation, likely because task-oriented dialogue demands high accuracy and strict adherence, while LLMs prioritize naturalness and diversity, often introducing redundancy or deviations from reference responses. In terms of efficiency, our method achieves the lowest latency among all approaches involving LLMs, with an average inference time of only 11.13 seconds. This is just about 0.2 seconds slower than the pure LM-based model (10.89 seconds), while still substantially faster than other strategies involving LLMs (e.g., 23.67 seconds for LLM coarse-to-fine retrieval). These findings demonstrate that our approach not only maintains high efficiency but also effectively leverages the advantages of LLMs. Overall, the coordinated fine-to-coarse retrieval with LM-based generation yields the most balanced and effective results in both quality and efficiency.

### 4.9 IMPACT OF LARGE LANGUAGE MODEL ON FINE-TO-COARSE RETRIEVAL AND RE-THINKING

To investigate the impact of different LLMs on fine-to-coarse retrieval and the Re-Thinker within the SMARTS framework, we evaluated five models on the MultiWOZ dataset: GPT-4o, GPT-4o-mini, Llama2-7B, Llama3-8B Dubey et al. (2024), and Deepseek-R1-Distill-Qwen (7B) Guo et al. (2025). The evaluation was conducted using entity-related metrics, including Entity F1, Entity Precision, and

Entity Recall. The experimental results are illustrated in Figures 3 and 4. Specifically, Llama2-7B demonstrated the weakest overall performance, with a particularly large gap compared to other models in fine-to-coarse retrieval. In contrast, GPT-4o and GPT-4o-mini achieved highly similar results across all three metrics under fine-to-coarse retrieval, suggesting that reducing model size did not substantially impair performance. Within the Re-Thinker, GPT-4o achieved the best results in terms of Entity F1 and Entity Precision, Llama3-8B excelled in Entity Recall, while Llama2-7B continued to lag behind across all measures. In summary, the findings indicate that within the SMARTS framework, model performance in fine-to-coarse retrieval and the Re-Thinker does not solely depend on model scale. Instead, it is more closely tied to the refinement of semantic representation and the stability of reasoning processes. In particular, stronger semantic encoding supports more accurate retrieval, while robust reasoning and self-correction capabilities enhance the effectiveness of the Re-Thinker. Therefore, improving semantic alignment and self-refinement abilities is more effective for boosting performance on complex tasks than merely increasing model size.

## 4.10 PERFORMANCE COMPARISON ON LARGE-SCALE KNOWLEDGE BASE

Previous methods are typically trained and evaluated on standardized-scale knowledge bases. However, in real-world scenarios, knowledge is retrieved from large-scale knowledge bases. To simulate such scenarios, following previous work Ding et al. (2024); Wan et al. (2023), we aggregate all knowledge records from the databases provided from the CamRest and MWOZ datasets to construct a unified large-scale dataset to evaluate the performance of the methods. The results are shown in Table 5. The results reveal that the performance of existing systems deteriorates when using the large-scale knowledge base

Table 5: The results of SMART and baseline models with a large-scale knowledge base on the MWOZ and CamRest datasets respectively.

| Method | CamRest | | MWOZ | |
| --- | --- | --- | --- | --- |
| | BLEU | Entity F1 | BLEU | Entity F1 |
| DF-Net | - | - | 6.45 | 27.31 |
| EER | 20.61 | 57.59 | 11.60 | 31.86 |
| FG2Seq | 19.20 | 59.35 | 10.74 | 33.68 |
| CDNet | 16.50 | 63.60 | 10.90 | 31.40 |
| Q-TOD | 21.44 | 63.88 | _16.67_ | 47.13 |
| MAKER | _26.19_ | _72.09_ | 16.25 | _50.87_ |
| Ours | **26.56** | **77.17** | **17.41** | **56.45** |

compared to their performance on standardized-scale knowledge bases. However, our model outperforms the baselines under the large-scale knowledge base setting. This improvement can be attributed to our fine-to-coarse knowledge retrieval mechanism and re-think mechanism, which enable it to retrieve more precise knowledge and relevant attributes during response generation.

## 4.11 HUMAN EVALUATION

We conduct a human evaluation by distributing an online survey to 50 university volunteers. Our method, SMART, is compared with two previous state-of-the-art (SOTA) models, MAKER and Uni-TOD. The evaluation assesses response naturalness, correctness, and task completion using 50 dialogue samples from the MWOZ dataset. The results, presented in Table 6, show that SMART

Table 6: The result of human evaluation on MWOZ dataset with stateful configuration.

| Model | Naturalness | Correctness | Task Completion |
| --- | --- | --- | --- |
| IEM | 3.9 | 3.7 | 3.8 |
| MAKER | 3.8 | 4.0 | 3.9 |
| Uni-TOD | 3.9 | 3.8 | 3.9 |
| SMART | **4.1** | **4.4** | **4.3** |

consistently achieves the highest scores across all metrics. These findings suggest that SMART integrates a more effective mechanism for generating human-like, contextually appropriate, and goal-oriented responses, making it a strong candidate for task-oriented dialogue systems.

## 5 CONCLUSION

In this paper, we propose SMART, a novel small-to-large model coordinated framework for task-oriented dialogue. SMART addresses the task-coupling limitations of traditional TOD systems by introducing a fine-to-coarse knowledge retrieval mechanism and a re-thinker. These designs enhance retrieval precision, reduce computational complexity, and improve response quality. By coordinating small and large language models, SMART effectively balances efficiency and informativeness. Extensive experiments on two public datasets demonstrate its superior knowledge retrieval accuracy and response generation capabilities, achieving effective improvements in Entity F1.

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

## A    THE USE OF LARGE LANGUAGE MODELS

In this paper, large language models were used exclusively for language polishing; they were not involved in any stage of paper conceptualization, structure design, or argument development.

## B    ABLATION STUDY ON CAMREST DATASET

To further validate the robustness of our approach, we additionally conduct an ablation study on the CamRest dataset. Table 7 reports the results. Specifically, we examine the impact of three key components: the fine-to-coarse knowledge retriever, the completed response generation task, and the re-thinker module.

The results show that removing any of these components consistently leads to performance degradation across most evaluation metrics. In particular, removing the fine-to-coarse knowledge retriever yields the largest drop in BLEU score and entity accuracy, highlighting its crucial role in ensuring precise knowledge selection. Similarly, excluding the completed response generation task reduces both precision and recall of entities, indicating that this task contributes to generating more informative and complete responses. Finally, removing the re-thinker module also results in noticeable performance decline, demonstrating the effectiveness of iterative refinement in improving overall system quality.

Overall, the ablation results on CamRest are largely consistent with those observed on the Multi-WOZ dataset, further confirming the generalizability and necessity of each component in our framework.

Table 7: Ablation study on the CamRest dataset.

| Method | BLEU | Entity | Entity Precision | Entity Recall | Entity Acc. |
|---|---|---|---|---|---|
| Ours | 27.21 | 79.72 | 79.92 | 79.53 | 54.43 |
| w/o fine-to-coarse knowledge retriever | 25.33 | 78.01 | 77.90 | 78.13 | 51.21 |
| w/o completed response generation task | 25.79 | 78.54 | 78.11 | 78.98 | 52.85 |
| w/o re-thinker | 26.71 | 79.15 | 79.40 | 78.90 | 53.91 |

## C    THE OVERALL INFERENCE PHASE

In the inference phase in SMART, the fine-tuned response style generator, with its compact parameter size, rapidly produces an initial style-only response $\hat{p}_i$. Attribute slots is extracted from $\hat{p}_i$ to form a filtered knowledge base $K'$, excluding irrelevant attributes. The fine-to-coarse knowledge retriever performs top-1 knowledge retrieval using $K'$ and dialogue context, filling attribute values into $\hat{p}_i$ to create a completed response $\hat{s}_i$. The re-thinker evaluates $\hat{s}_i$ for quality and contextual appropriateness. If $\hat{s}_i$ meets quality standards, it serves as the final response; otherwise, a re-generation phase employs an LLM to produce a refined, high-quality response. This mechanism ensures SMART balances speed and accuracy. The overall pseudo code of the SMART inference phase is presented in Algorithm 1.

---

**Algorithm 1:** Pseudo code of SMART inference

---

**Input:** $i$-th turn Dialogue context $d_i$, Knowledge base $K$

**Output:** Response from $i$-th turn dialogue $\hat{s}_i$

1   $\hat{p}_i, A'_i \leftarrow \text{LM}(d_i^{trn})$ // Obtain a style-only response and attribute slots

2   $K' \leftarrow \text{Filter}(K, A'_i)$ // Obtain a attribute filtered knowledge base according to the attribute slots

3   $e_i^{top} \leftarrow \text{LLM}(d_i, K', P^r)$ // Obtain the top-1 knowledge record by a dialogue context and a attribute filtered knowledge base

4   $\hat{s}_i \leftarrow \text{Fill}(e_i^{top}, \hat{p}_i)$ // Fill the attribute values into the style-only response and obtain a completed response.

5   $r \leftarrow \text{Re-Thinker}(\hat{s}_i, d_i)$ // Determine whether the response is logical based on the dialogue context and return a context-sensitive boolean value $r$.

6   **if** $r$ *is No* **then**

7     $\hat{s}_i \leftarrow \text{LLM}(d_i, \hat{s}_i, e_i^{top}, P^g)$ // Re-generate a response as the final completed response.

---

