# OpenReview forum: "SMARTS: A Small-to-large Model Coordinated Retrieval and Response Framework for Task-Oriented Dialogue"
_ICLR.cc/2026/Conference — ICLR 2026 Conference Withdrawn Submission_

### Official Review · Reviewer_hdBk · 2025-10-21

**Soundness:** 2
**Presentation:** 3
**Contribution:** 1
**Rating:** 2
**Confidence:** 4

**Summary:**

The authors focus on the problem of incorporating responses in task-oriented dialog (TOD) systems. They state that the issue with current TOD systems arent able to injecting knowledge effectively in their responses either because: the LLM has to do both response generation and knowledge retrieval which makes the task harder or if there is a retrieval mechanism incorrect knowledge can be retrieved.

The authors approach is to have Small Language Model (SLM) generate style-only responses containing placeholder slots for attribute and a LLM to retrieve the most relevant knowledge records by filtering on attribute slots. The values retrieved for those particular slots are filled in for the final response. Additionally there is a separate LLM or Re-Thinker that determines if the final response is contextually relevant if it is not the Re-Thinker model regenerates the response. The response-style generator is finetuned on two task-oriented datasets, the knowledge retriever and re-thinker model are not.

The authors show that this performs baselines which use the previous methods mentioned above on metrics such as BLEU (4-gram), Entity F1, Entity precision, Entity recall, and Entity accuracy. The authors perform an ablation study to show each part of their system is valuable. They also compare their method versus just asking an LLM to generate a response with retrieved knowledge and show it is better than this approach.

The contributions as stated in the paper is: 1) a method that leverages a SLM to generate style-only responses to be filled in and 2) a more accurate knowledge retrieval mechanism.

**Strengths:**

1) The authors certainly describe their method in a clear manner and run a fair number of ablation studies to show the different components of their work.

2) Additionally they use an extensive set of baselines and even compare against very powerful LLMs.

**Weaknesses:**

There are a couple claims in the paper that I think can be clarified more:

1) It is mentioned in Table 3 that "Although LLMs have strong reasoning ability, directly using them for response generation without finetuning often results in verbose outputs containing irrelevant or excessive attributes." While I understand that current LLMs do generate long outputs the responses could still be useful and it would be hard to determine if it's worse or better for a user without human evaluation. Also if you wanted couldn't you add in the prompt to make sure the responses are short or ask it to not use excessive prompts? What was the prompt used? Is it the same prompt used in the re-thinker?

2) In what cases would the response not be appropriate? Since you are retrieving the exact attributes needed along with their slots. Also how much of the knowledge gets filtered out during your retrieval on average.

3) One of the novelties mentioned in the paper is using attribute slots to filter out knowledge records, but it seems like MAKER already does this so what's the difference?

**Questions:**

1) Why was SAGE not used as a baseline?

2) What are the SOTA numbers on the two datasets you used? I'm asking because I'd like to know if the methods used for those are different than yours or is your approach the SOTA?

3) More of a suggestion: Put link to Table 1 in Section 4.5. Also are the results statistically significant?

4) What is LM vs LLM? Do you mean SLM?

---

> ### Author Response · Authors · 2025-11-23
>
> ***It is mentioned in Table 3 that "Although LLMs have strong reasoning ability, directly using them for response generation without finetuning often results in verbose outputs containing irrelevant or excessive attributes." While I understand that current LLMs do generate long outputs the responses could still be useful and it would be hard to determine if it's worse or better for a user without human evaluation. Also if you wanted couldn't you add in the prompt to make sure the responses are short or ask it to not use excessive prompts? What was the prompt used? Is it the same prompt used in the re-thinker?***
>
> Response:
> Because the knowledge base typically contains multiple entities, each with several attributes, an untuned LLM often tends to reproduce attributes from multiple entities together in its response. This leads to information mixing and prevents users from receiving answers targeted to the specific task, ultimately resulting in lower BLEU and Entity F1 scores. This is a
> consistent and reproducible behavior of LLMs in task-oriented dialogue settings. Modifying the prompt cannot fully prevent this issue, as LLMs still exhibit a tendency to enumerate all available attributes when presented with multi-attribute records. In addition, the prompt used for this baseline is not the same as the one used for the Re-thinker module. Here, the prompt is exclusively for instructing the LLM to directly generate a response based on the dialogue context and knowledge base.
>
> ***In what cases would the response not be appropriate? Since you are retrieving the exact attributes needed along with their slots. Also how much of the knowledge gets filtered out during your retrieval on average.***
>
> Response: In the SMART framework, a filled response may still be judged as “inappropriate” by the Re-thinker even when the correct attribute-value pairs have been retrieved. This is because the Re-thinker’s criteria extend beyond simply checking whether the response contains the correct attributes, it also verifies whether the response omits any required slots specified by the SLM, whether it introduces additional inferential content that were not retrieved, and whether its phrasing deviates from the user’s intent or task-oriented dialogue conventions. When such issues occur, the Re-thinker triggers a controlled regeneration step using the LLM to ensure that the final response strictly adheres to the retrieved attributes and remains concise and task-compliant.
> Regarding the extent of filtering, since the SLM typically produces only 2–3 slots while each knowledge record usually contains 5–10 attributes, our filtering step removes approximately 60%–70% of irrelevant attributes on average before retrieval. This substantially reduces the comparison space for the LLM and mitigates the risk of introducing unnecessary attributes during generation.
>
> ***One of the novelties mentioned in the paper is using attribute slots to filter out knowledge records, but it seems like MAKER already does this so what's the difference?***
>
> Response:
> MAKER indeed includes both entity selection and attribute selection, but its attribute filtering mechanism is fundamentally different from the attribute-slot filtering proposed in our work. MAKER’s attribute selector is trained using cross-attention distillation and pseudo-labels to produce attribute-importance scores. This results in a soft selection mechanism in which attributes are not removed from the record. Instead, they are retained with different weights in subsequent matching, and the filtering is guided by the model’s learned semantic relevance.
> In contrast, SMART’s filtering is entirely driven by explicit attribute slots generated by a small language model. These slots correspond to concrete fields in the structured knowledge base (e.g., price, area). We perform hard, field-level filtering on each knowledge record by removing all attributes not present in the slot set, and only the pruned records are passed to the frozen LLM for fine-to-coarse ranking. As a result, SMART’s filtering is controllable, interpretable, and structurally grounded, leading to substantially improved entity consistency and accuracy.
>
> ***What is LM vs LLM? Do you mean SLM?***
>
> Response: In the current version of the paper, "LM" refers to SLM. In the revised manuscript, we will unify the terminology by replacing all occurrences of "LM" with "SLM" to avoid ambiguity.

---

### Official Review · Reviewer_ewcP · 2025-10-27

**Soundness:** 2
**Presentation:** 3
**Contribution:** 2
**Rating:** 2
**Confidence:** 3

**Summary:**

This work proposes a small-to-large model for task-oriented dialogue generation.

The overall idea of  `SMART`  is a simple two-stage pipeline:

- First, it uses a small language model (Flan-T5) to generate a prototype dialogue response ( style response), which leaves the knowledge entity as some placeholders.
- Then, it uses a large language model (Llama3-8B) to fill the entity placeholders.

Experiments on two datasets have shown the effectiveness of the proposed method in Entity-Related metrics.

**Strengths:**

1. The presentation of this work is easy to follow.

2. `SMART` has chosen a different paradigm compared to the most related works.

3. Compared to the baselines selected by the authors, the performance of  `SMART` is not bad.

**Weaknesses:**

1. Line 127:  A lot of pipeline-based works use different models to retrieve knowledge and generate response, instead of a single model. In the era of RAG, a lot of works use different modules in different stages, then now there are already a lot of dense retrievers.

2. The novelty of this work is limited.  The core idea of this work is to first generate a skeleton response, which was populated about 5 years ago (just like `Skeleton-to-Response: Dialogue Generation Guided by Retrieval Memory`).  Besides, I also can't find other innovations that are novel enough.

3. According to the experimental results, I think the experimental setting is specifically optimized for non-LLM models.  Many metrics are not suitable in the era of LLMs, especially reference-aware metrics.  Thus, the comparison results between the well-trained small language model and the instructional large language model are somewhat meaningless in this paper.

4. Lack the evaluation of time complexity.

5. Lack the discussion/comparison of using powerful dense retrievers.

6. The proposed paradigm still requires a lot of  `Supervised Fine-Tuning` processes when constructing the fundamental modules, which is outdated in the era of LLM. Only a limited set of scenarios can benefit from this work.

**Questions:**

1. Line 37-43:  This paper cited some works that use the entire knowledge base to generate the dialogue. It is strange because most works will have an explicit knowledge retrieval process (such as Figure 1b). The full text length of most knowledge bases is often far beyond the LLM input limitation. So, I am wondering about the practical scenario of this paradigm.

---

> ### Author Response · Authors · 2025-11-23
>
> ***W2: The novelty of this work is limited. The core idea of this work is to first generate a skeleton response, which was populated about 5 years ago (just like Skeleton-to-Response: Dialogue Generation Guided by Retrieval Memory). Besides, I also can't find other innovations that are novel enough.***
>
> Response: There are two misunderstandings in equating our method with the traditional “generate a skeleton and then fill it” paradigm. First, SMART is designed for task-oriented dialogue grounded in a structured knowledge base. Its core objective is not to extract word-level skeletons from retrieved responses to facilitate open-domain generation. Instead, SMART employs a small language model to explicitly generate a style-only response with attribute slots, which semantically identifies the required attribute dimensions. These slots are then used to perform attribute-level filtering over the knowledge base, after which a frozen large language model conducts semantic, record-level selection and a subsequent verification step. This workflow is fundamentally different from skeleton-to-response approaches. Second, SMART introduces several design components that extend beyond the traditional skeleton paradigm, including a completed-response auxiliary training objective, an Adapter-based SLM architecture, and an LLM-driven re-thinker for response validation and regeneration. These are not merely architectural combinations, they are specifically developed to address challenges unique to task-oriented dialogue, such as accurate attribute grounding and robustness under large-scale knowledge bases. Our ablation studies and large-knowledge base experiments further demonstrate that each component contributes substantially to the improvements in entity-level accuracy and overall system performance, as reported in the experimental section.
>
> ***W3: According to the experimental results, I think the experimental setting is specifically optimized for non-LLM models. Many metrics are not suitable in the era of LLMs, especially reference-aware metrics. Thus, the comparison results between the well-trained small language model and the instructional large language model are somewhat meaningless in this paper.***
>
> Response: The experimental setup in this paper is not intentionally biased toward non-LLM models, rather, it follows established evaluation practices in the task-oriented dialogue community. The fundamental objective of task-oriented dialogue systems is to generate natural and accurate responses based on a dialogue context and a knowledge base. Therefore, reference-aware metrics such as BLEU, Entity F1, and Entity Accuracy remain widely recognized and necessary for assessing task completion and knowledge grounding. These metrics are not designed to favor small language models, but are intrinsic to the TOD task itself. Moreover, the goal of our experiments is not to show that small language models outperform LLMs, but to verify whether SMART’s task-decoupled framework is more effective than directly relying on LLMs for generation. The results indicate that although LLMs possess stronger language modeling capabilities, they often produce redundant content or attribute errors in TOD scenarios, leading to degraded performance. In contrast, SMART's fine-to-coarse retrieval strategy and re-thinker alleviate these issues. Thus, the experimental results demonstrate not only the validity of our evaluation setup, but also the necessity of establishing a structured, controllable, and verifiable cooperation framework between small and large models in TOD, rather than applying special optimization for non-LLM models.
>
> ***W6: The proposed paradigm still requires a lot of Supervised Fine-Tuning processes when constructing the fundamental modules, which is outdated in the era of LLM. Only a limited set of scenarios can benefit from this work.***
>
> We would like to emphasize that SMART does not revert to a heavy supervised fine-tuning (SFT) paradigm, nor does it conflict with the development trend of LLMs. In SMART, the only component that requires fine-tuning is the lightweight small language model (SLM), whose role is to extract attribute slots and control the response style. The other parts of the framework (including attribute-level filtering, record-level selection, and re-thinker) are performed entirely by a frozen LLM, without any parameter updates to the LLM. Compared with large-scale SFT or RLHF applied directly to an LLM, SMART offers lower training cost and greater adaptability across domains. More importantly, task-oriented dialogue systems must generate structured, controllable, and verifiable attribute-level content, which cannot yet be reliably achieved through zero-shot LLM generation alone. Therefore, SMART is not limited to a narrow set of scenarios, instead, it provides a scalable paradigm for maintaining control and reliability in the era of LLMs.

---

> ### Author Response · Authors · 2025-11-23
>
> ***Line 37-43: This paper cited some works that use the entire knowledge base to generate the dialogue. It is strange because most works will have an explicit knowledge retrieval process (such as Figure 1b). The full text length of most knowledge bases is often far beyond the LLM input limitation. So, I am wondering about the practical scenario of this paradigm.***
>
> Response: In citing prior work that generates responses directly based on the full knowledge base, our goal is to illustrate that task-oriented dialogue research indeed includes methods that do not rely on explicit retrieval, instead feeding the entire knowledge base into the model or incorporating it into model parameters. Such methods may linearize the knowledge base and concatenate it with the dialogue context, or allow the model to learn an implicit knowledge distribution from the full knowledge base during training. These methods do not literally input every record individually, rather, they rely on structured encoding, schema expansion, or parameter internalization so that the model can operate without explicit retrieval at inference time. We reference these studies to highlight that although some work attempts to reduce the retrieval step, such strategies encounter practical issues in real-world TOD scenarios, including input length constraints, noisy attribute information, and difficulty ensuring accurate entity grounding. This motivates the design of SMART. Instead of following full- knowledge base input paradigms, SMART employs a small language model to generate attribute slots and perform fine-grained attribute-level filtering, effectively narrowing the knowledge base to a highly relevant subset. A frozen LLM then conducts semantic, record-level selection and verification over this reduced search space.

---

### Official Review · Reviewer_F4Ww · 2025-10-31

**Soundness:** 1
**Presentation:** 2
**Contribution:** 2
**Rating:** 2
**Confidence:** 4

**Summary:**

This paper introduces SMART, a modular framework for task-oriented dialogue (TOD) that decouples response style generation (handled by a small LM) from knowledge retrieval (handled by a frozen LLM). A fine-to-coarse retrieval mechanism is proposed that uses attribute slots from style-only responses to filter and select the top-1 relevant knowledge record. A Re-thinker module is further employed to verify contextual relevance and regenerate responses if necessary. Experiments on CamRest and MultiWOZ show consistent improvements over strong baselines across multiple metrics.

**Strengths:**

- The method proposed represents an improvement over the chosen baseline.
- The paper reads smoothly, with no obvious grammatical issues.

**Weaknesses:**

The weaknesses of the article are summarized below; see the **Questions** for specific corresponding issues.
- Limited Innovation: There are many works that decouple retrieval and generation, and there are also simpler and more efficient methods. The innovativeness of this paper is not very clear.
- The approach and motivation are quite confusing, lacking analysis and demonstration of why the proposed method works.
- The performance comparison is questionable: it cannot be ruled out that the gain obtained by this method may come from multiple LLM calls.

**Questions:**

### Typro
-  Line 184: an SLM -> a SLM。
-  Line 233: designed as -> is designed as

### Questions
- How does this method differ from previous paradigms that separated retrieval and generation, such as MAKER[1], SynctTOD[2]? For example, is the style generator used to predict relevant attributes? If so, why generate a template with slots? Can't we just generate the relevant attributes directly? If we generate a stylized sentence, will it limit the model's response capability? I don't think this method is fundamentally innovative compared to previous methods.
- Why can large, untrained language models replace dedicated retrieval components for ultra-large-scale retrieval? What is the inference latency of large models used for retrieval? Does the length of the text input to a large database exceed the understanding length of an LLM?
- Line 201 “such as assessing whether specific attribute values are appropriate in context.” How can we assess whether an attribute is appropriate across contexts? why the CE(ˆpi, pi) only can not do this?
- Where does the golden style-only response come from? Is it in the dataset?
- The authors introduce an auxiliary loss that trains the style generator (SLM) to predict the final response (i.e., with knowledge filled in), in addition to the style-only template. While the stated goal is to “enhance the model’s understanding of fine-grained knowledge usage,” this objective appears to contradict the core design principle of full decoupling between style and knowledge. Could the authors clarify how this supervision on the completed response does not reintroduce a coupling between generation and knowledge retrieval? And provide an ablation study about the influence of CE(ˆs′i, si). Since multi-task optimization loss is used, how can we ultimately guarantee the generation of style responses with slots?
- Is there a comparison and analysis between not generating style responses and generating attributes without them? Is there an explanation of why style responses are better, and what makes them reasonable?
- Rethinking the mechanism and LLM retrieval involves frequent calls to large models, which increases the response latency for task-oriented dialogues. Has the performance-latency ratio of the model been measured?
- Lack of comparison with related work, such as syncTOD [2], etc.
- On the claim of “fine-to-coarse” retrieval. The paper advertises a fine-to-coarse retrieval mechanism; however, the actual pipeline performs a single retrieval step over an attribute-filtered knowledge base. We do not observe a progressive widening of candidate scope.
- Insufficient per-component ablation study, such as slot accuracy: The quality of the slots Ai′ extracted by the SLM is not reported. Errors at this stage are irreversible and could mislead the entire retrieval. Please report the precision of slot extraction, and analyze how slot errors influence the generation quality.

[2] Synergizing In-context Learning with Hints for End-to-end Task-oriented Dialog Systems

---

> ### Author Response · Authors · 2025-11-23
>
> ***Q1: How does this method differ from previous paradigms that separated retrieval and generation, such as MAKER[1], SynctTOD[2]? For example, is the style generator used to predict relevant attributes? If so, why generate a template with slots? Can't we just generate the relevant attributes directly? If we generate a stylized sentence, will it limit the model's response capability? I don't think this method is fundamentally innovative compared to previous methods.***
>
> Response:
>
> 1) The key difference between our method and prior work such as MAKER and SyncTOD lies in the deeper decoupling of responsibilities: we not only separate retrieval and generation, but also assign style and slot-structure modeling to a small language model, while delegating semantic discrimination and entity retrieval to a frozen large language model, coupled with an attribute-slot-based fine-to-coarse retrieval mechanism. These two aspects are not present in existing approaches. Specifically, although MAKER performs multi-granularity retrieval through entity and attribute selection, its generation stage still requires selecting the most relevant knowledge from multiple related entities and attributes. In contrast, SMART directly fills slot values with the accurately retrieved knowledge. SyncTOD employs hints to guide exemplar selection and prompting for in-context learning, focusing on encouraging the LLM to produce stylistically aligned responses under few-shot settings. SMART differs fundamentally by letting the small language model learn response style while the frozen large language model performs precise retrieval, which yields more reliable entity accuracy and style control under large-scale knowledge bases.
>
> 2) The style generator does not merely predict attributes, rather, it generates a slot-based style response. This design preserves the clear structure and politeness conventions required in task-oriented dialogue, while also substantially reducing retrieval noise. The predicted slots explicitly determine which attributes should be compared, enabling the frozen LLM to evaluate only the coarse-filtered knowledge records and greatly reducing interference from irrelevant information.
>
> 3) Our SMART does not constrain the model’s response capability. The SLM controls only the response style and slot structure, while the factual content and final natural-language realization are produced through frozen-LLM retrieval followed by re-thinking verification or regeneration. As shown in our experiments, this design yields responses that are more stable than direct LLM generation, and achieves substantial improvements in both Entity F1 and BLEU.
>
> ***Q2: Why can large, untrained language models replace dedicated retrieval components for ultra-large-scale retrieval? What is the inference latency of large models used for retrieval? Does the length of the text input to a large database exceed the understanding length of an LLM?***
>
> Response:
>
> 1)	In SMART, the large language model does not perform full-database retrieval. Instead, conditioned on the style-only response and its attribute slots generated by the SLM, each knowledge record is filtered to retain only the attributes relevant to the current turn. Consequently, the large language model only needs to conduct semantic discrimination over a small set of short and filtered candidates, allowing it to achieve strong retrieval performance without any additional training.
>
> 2)	Because the input to the large language model is short, the latency for a single retrieval judgment using a frozen LLM is low. This is significantly lower than the cost of full-sentence generation by an LLM and does not involve any large-scale vector retrieval computation.
>
> 3)	Moreover, after slot-based filtering, each knowledge record is reduced to a compact subset of attribute fields, and the number of candidates passed to the LLM is small. Therefore, the total input remains well within the LLM’s context window, avoiding issues related to context length limitations.
>
> ***Q3: Line 201 “such as assessing whether specific attribute values are appropriate in context.” How can we assess whether an attribute is appropriate across contexts? why the CE(ˆpi, pi) only can not do this?***
>
> Response: CE loss can only supervise whether the generated template and slot structure are correct. It cannot ensure that the selected entity attributes are the most appropriate for the given dialogue context. To address this limitation, we introduce an auxiliary completed-response prediction task together with a completed-response adapter in the decoder. This task compels the model to generate full responses during training, enabling it to learn the contextual patterns in which attributes are used and thereby acquire finer-grained knowledge utilization capabilities beyond what template-only supervision can provide.

---

> > ### Comment · Reviewer_F4Ww · 2025-11-26
> >
> > As the authors have so far addressed only part of my concerns, I **maintain** my current score for the moment.
> > Regarding Q2, could the authors please provide a **quantitative** analysis？

---

> > > ### Author Response · Authors · 2025-11-28
> > >
> > > For Q2, we think that Section 4.8 and Table 4 (Lines 418–421) have discussed the latency issue to some extent by comparing the efficiency.

---

> ### Author Response · Authors · 2025-11-23
>
> ***Q4: Where does the golden style-only response come from? Is it in the dataset?***
>
> Response: The golden style-only response is not directly provided in the original dataset. Instead, we derive it from each full human-written response by applying a matching algorithm that replaces concrete attribute values with their corresponding attribute slots.
>
> ***Q5: The authors introduce an auxiliary loss that trains the style generator (SLM) to predict the final response (i.e., with knowledge filled in), in addition to the style-only template. While the stated goal is to “enhance the model’s understanding of fine-grained knowledge usage,” this objective appears to contradict the core design principle of full decoupling between style and knowledge. Could the authors clarify how this supervision on the completed response does not reintroduce a coupling between generation and knowledge retrieval? And provide an ablation study about the influence of CE(ˆs′i, si). Since multi-task optimization loss is used, how can we ultimately guarantee the generation of style responses with slots?***
>
> Response: The completed-response auxiliary task does not compromise the core design principle of fully decoupling style from knowledge. This task is connected to the decoder only through a lightweight adapter, which is entirely disabled during inference. Moreover, its objective is not to make the SLM predict specific entities, rather, it exposes the model during training to the structure of full responses and the common patterns of attribute co-occurrence. This helps the SLM learn which slots are typically needed and how slots interact within the response structure, without learning any actual knowledge content. Thus, the task only provides structural supervision, and does not reintroduce coupling with knowledge retrieval. The selection of entities remains fully handled by the subsequent fine-to-coarse LLM-based retrieval module.
>
> ***Q9: On the claim of “fine-to-coarse” retrieval. The paper advertises a fine-to-coarse retrieval mechanism; however, the actual pipeline performs a single retrieval step over an attribute-filtered knowledge base. We do not observe a progressive widening of candidate scope.***
>
> Response: Your interpretation of fine-to-coarse as a progressive expansion of the candidate set differs from how the term is defined in our paper. In SMART, fine-to-coarse refers to a progression in the granularity of retrieval signals, rather than a multi-stage widening of candidates. Specifically, the attribute slots predicted by the SLM determine which attribute dimensions are actually relevant to the current dialogue turn. This step performs fine-grained, attribute-level filtering over the original knowledge base, compressing each record to only the fields strongly aligned with the current context. The frozen LLM then performs coarse-grained, entity-level semantic discrimination over this tightly filtered candidate set, directly selecting the best-matching record. Thus, the fine→coarse process denotes a transition from attribute-level fine filtering to entity-level semantic matching, not a retrieval pipeline in which the candidate pool gradually expands. This mechanism therefore operates by first precisely narrowing the search space and then using the large language model for coarse semantic judgment, which substantially improves the accuracy and efficiency of entity selection.

---

### Official Review · Reviewer_LxKw · 2025-11-01

**Soundness:** 2
**Presentation:** 2
**Contribution:** 2
**Rating:** 2
**Confidence:** 4

**Summary:**

This paper introduces a task-oriented dialogue framework named SMART, which addresses the coupling issue of knowledge retrieval and response generation in traditional TOD systems through task decoupling. The SMART framework consists of three main components: 1) a style generator, which generates stylized responses based on the dialogue context; 2) a fine-grained knowledge retriever, which utilizes large models for fine-grained knowledge retrieval; and 3) a reflection module, which re-validates the generated responses to ensure their quality and accuracy. Experimental results on two public datasets, Camrest and MultiWOZ, show that the SMART framework improves the entity F1 score by an average of 5.79% compared to other methods. Furthermore, the SMART framework boasts lower computational costs and higher response quality.

**Strengths:**

1. The authors conduct thorough experiments using two public datasets (Camrest and MultiWOZ), evaluating their proposed method against various baselines and ablation studies. They demonstrate significant improvements in entity F1 scores and discuss the efficiency and accuracy trade-offs of their approach.

2. The paper is well-structured and clearly explains the motivation behind the proposed method, providing detailed descriptions of the SMART architecture and key mechanisms such as attribute filtering, top-1 knowledge retrieval, and reflection module. The experimental setup and results are presented in a logical manner, allowing readers to easily understand the findings.

**Weaknesses:**

1. Novelty. The approach of removing keywords for style generation is not uncommon, and the overall pipeline is a conventional practice, which may not meet the high standards of ICLR.

2. Dataset Dependency.
(1)The evaluation of the SMART framework is primarily based on two publicly available datasets, Camrest and MultiWOZ.
(2)The performance on out-of-domain (OOD) datasets has not been tested. While these datasets are widely used in the TOD research community, future work could explore the generalizability of the proposed method across different domains and datasets.

**Questions:**

1. Why is the output from the direct LLM almost unusable? Does this imply that existing RAG systems are unreliable?

2. Why does the Finetune-LLM perform worse than the base LM in generation?

---

> ### Author Response · Authors · 2025-11-15
>
> ***W1**: Novelty. The approach of removing keywords for style generation is not uncommon, and the overall pipeline is a conventional practice, which may not meet the high standards of ICLR.*
>
> ---
>
> Response: Our work is not merely about "removing keywords for style generation", **but rather proposes a small-to-large model collaborative framework for task decoupling**. In SMART framework, a small language model generates style-only responses, while a frozen large language model serves as a fine-to-coarse knowledge retriever to retrieve the top-1 most relevant knowledge record. A re-thinker is further introduced to verify contextual consistency and knowledge accuracy. **This style-knowledge decoupling with small-large model collaboration mechanism differs from traditional end-to-end or single-model pipelines.** Experimental results also demonstrate that our method improves both knowledge accuracy (Entity F1) and response naturalness (BLEU).
>
> ***W2**: Dataset Dependency. (1)The evaluation of the SMART framework is primarily based on two publicly available datasets, Camrest and MultiWOZ. (2)The performance on out-of-domain (OOD) datasets has not been tested. While these datasets are widely used in the TOD research community, future work could explore the generalizability of the proposed method across different domains and datasets.*
>
> ---
>
> Response: Camrest and MultiWOZ are two of the most representative benchmark datasets in task-oriented dialogue research. Specifically, **Camrest focuses on the restaurant reservation domain**, making it suitable for evaluating knowledge retrieval and response generation in a single-domain setting. In contrast, **MultiWOZ covers five distinct domains (including train, hotel, restaurant, taxi, and attraction)** providing a comprehensive benchmark for assessing a model's generalization and robustness across multiple domains. More importantly, the design and training process of SMART are domain-agnostic, giving the framework inherent transferability across different dialogue domains.
>
> ***Q1**: Why is the output from the direct LLM almost unusable? Does this imply that existing RAG systems are unreliable?*
>
> ---
>
> Response: The statement that "the output from the direct LLM almost unusable" specifically refers to task-oriented dialogue scenarios without specific tuning, and does not indicate that RAG systems are unreliable in general. **Task-oriented dialogues require responses that are concise, natural, and aligned with user intent, whereas untuned LLMs often produce overly verbose outputs containing irrelevant attributes, leading to decreases in BLEU and Entity F1 scores.** In contrast, the style-only small language model can consistently generate controllable and stylistically coherent responses by training, which are then complemented with retrieved knowledge by the large language model, resulting in higher-quality task-oriented response.
>
> ***Q2**: Why does the Finetune-LLM perform worse than the base LM in generation?*
>
> ---
>
> Response: If you are talking about Table 4, there might be a misunderstanding. **The LLM in our framework is not fine-tuned, it performs generation under a zero-shot setting**. For better clarification, we will add an additional experiment to illustrate the performance of a fine-tuned LLM. The main reason why the LLM performs worse than the LM in generation is explained in lines 415–418 of the "Coordinated Strategy of Large-Small Language Model Framework" section.

---

### Note · Authors · 2026-01-02

I have read and agree with the venue's withdrawal policy on behalf of myself and my co-authors.